# GradMax: Growing Neural Networks using Gradient Information

**Utku Evci, Bart van Merriënboer, Thomas Unterthiner,**
**Max Vladymyrov, Fabian Pedregosa**
Google Research, Brain Team
{evcu,bartvm,unterthiner,mxv,pedregosa}@google.com

## Abstract

The architecture and the parameters of neural networks are often optimized independently, which requires costly retraining of the parameters whenever the architecture is modified. In this work we instead focus on growing the architecture without requiring costly retraining. We present a method that adds new neurons during training without impacting what is already learned, while improving the training dynamics. We achieve the latter by maximizing the gradients of the new weights and efficiently find the optimal initialization by means of the singular value decomposition (SVD). We call this technique Gradient Maximizing Growth (GradMax) and demonstrate its effectiveness in variety of vision tasks and architectures[1].

## 1 Introduction

The architecture of deep learning models influences a model's inductive biases and has been shown to have a crucial effect on both the training speed and generalization (dÁscoli et al., 2019; Neyshabur, 2020). Searching for the best architecture for a given task is an active research area with diverse approaches, including neural architecture search (NAS) (Elsken et al., 2019), pruning (Liu et al., 2018), and evolutionary algorithms (Stanley & Miikkulainen, 2002). Most of these approaches are costly, as they require large search spaces or large architectures to start with. In this work we consider an alternative approach: Can we start with a small network and learn an efficient architecture without ever needing a large network or exceeding the size of our final architecture?

The idea of incrementally increasing the size of a model has been used in many settings such as boosting (Friedman, 2001), continual learning (Rusu et al., 2016), architecture search (Elsken et al., 2017; Cortes et al., 2017), optimization (Fukumizu & Amari, 2000; Caccia et al., 2022), and reinforcement learning (Berner et al., 2019). Despite the wide range of applications of growing neural networks, the initialization of newly grown neurons is rarely studied. Existing work on growing neural networks either adds new neurons randomly (Chen et al., 2016; Berner et al., 2019) or chooses them with the aim of decreasing the training loss (Bengio et al., 2006; Liu et al., 2019; Wu et al., 2020a). In this work we take a different approach. *Instead of improving the training objective immediately through growing, we focus on improving the subsequent training dynamics*. As we will show, this has a longer-lasting effect than the greedy approach of improving loss during growing.

Our main contribution is a new growing method that maximizes the gradient norm of newly added neurons, which we call GradMax. We show that this gradient maximization problem can be solved in a closed form using the singular value decomposition (SVD) and initializing the new neurons using the top $k$ singular vectors. We show that the increased gradient norm persists during future steps, which in return yields faster training.

## 2 Growing neural networks

Neural network learning is often formalized as the following optimization problem:

$$\arg\min_{f \in S} \mathbb{E}_{(\boldsymbol{x}, \boldsymbol{y}) \sim D} \left[ L(f(\boldsymbol{x}), \boldsymbol{y}) \right] , \tag{1}$$

---

[1]We open source our code at https://github.com/google-research/growneuron.

where $S$ is a set of neural networks, $L$ is a loss function and $D$ is a data set consisting of inputs $\boldsymbol{x}$ and outputs $\boldsymbol{y}$). Most often the set $S$ is constrained to a single architecture (e.g., ResNet-18 (Zagoruyko & Komodakis, 2016)) and only the parameters of the model are learned. However, hand-crafted architectures are often suboptimal and various approaches aim to optimize the architecture itself together with its parameters.

Methods such as random search (Bergstra & Bengio, 2012), NAS, and pruning search a larger space that contains a variety of architectures. Growing neural networks has the same goal: It learns the architecture and weights jointly. It aims to achieve this by incrementally increasing the size of the model. Since this approach never requires training a large model from scratch it needs less memory and compute than pruning methods (Yuan et al., 2021). Reducing the cost of finding efficient architectures is important given the ever-growing size of architectures and the accompanying increase in environmental footprint (Thompson et al., 2021).

**Growing neural networks: When, where and how?** Algorithms for growing neural networks start training with a smaller *seed architecture*. Then over the course of the training new neurons are added to the seed architecture, either increasing the width of the existing layers or creating new layers. Algorithms for growing neural networks should address the following questions:

1. **When** to add new neurons? For instance, some methods (Liu et al., 2019; Kilcher et al., 2019) require the training loss to plateau before growing, whereas others grow using a predefined schedule.

2. **Where** to add new capacity? We can add new neurons to the existing layers or create new layers among the existing ones.

3. **How** to initialize the new capacity?

In this work we mainly focus on the question of **how** and introduce a new initialization method for the new neurons. Our approach can also be used to guide when and where to grow new neurons. However, in order to make our comparison with other initialization methods fair we keep the growing schedule (**where** and **when**) fixed.

**How to grow new neurons?** Suppose that during training we would like to grow $k$ new neurons at layer $\ell$ as depicted in Figure 1. After the growth, new neurons are appended to the existing weight matrices as follows:

$$W_\ell^+ = \begin{bmatrix} W_\ell \\ W_\ell^{\text{new}} \end{bmatrix} \tag{2}$$

$$W_{\ell+1}^+ = \begin{bmatrix} W_{\ell+1} & W_{\ell+1}^{\text{new}} \end{bmatrix} . \tag{3}$$

A desirable property of growing algorithms is to preserve the information that the network has learned. Therefore the initialization of new neurons should ensure our neural network has the same outputs before and after growth.

We describe two main approaches to growing: splitting and adding. **Splitting** (Chen et al., 2016; Liu et al., 2019) duplicates existing neurons and adjusts outgoing weights so that the output in the next layer is unchanged. However, splitting has some limitations: (1) It creates neurons with the same weights and small changes required to break symmetry; (2) It can't be used for growing new layers as it requires existing neurons to begin with.

Another line of work focuses on **adding** new neurons while ensuring that the output does not change by setting either $W_\ell^{\text{new}}$ or $W_{\ell+1}^{\text{new}}$ to zero. This approach doesn't have the limitations mentioned above and often results in better performance (Wu et al., 2020a). Below we summarize two methods that use this approach:

**Random** Berner et al. (2019) initialize new neurons randomly when growing new layers.

**Firefly** Wu et al. (2020a) combine splitting and adding random neurons. Random neurons are trained for a few steps to reduce loss directly and the most promising neurons are selected for growing.

Previous work focused either on solely keeping the output of the neural network the same (Random) or on immediately reducing the training objective (Firefly). In this work we take a different approach

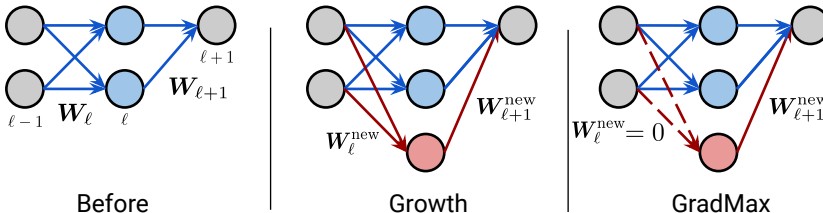

Figure 1: Schematic view of the GradMax algorithm. Growing new neurons requires initializing incoming ($\boldsymbol{W}_\ell^{\text{new}}$) and outgoing $\boldsymbol{W}_{\ell+1}^{\text{new}}$ weights for the new neuron. GradMax sets incoming weights to zero (dashed lines) in order to keep the output unchanged, and initializes outgoing weights using SVD (equation 11). This maximizes the gradients on the incoming weights with the aim of accelerating training.

and focus on improving the training dynamics. We initialize the weights so that the network's output remains unchanged while trying to improve the training dynamics by maximizing the gradient norm. The hypothesis is that this accelerates training and leads to a better model in the long run.

## 3 GRADMAX

This section describes our main contribution: GradMax, a method that maximizes the gradient norm of the new weights. We constrain the norm of the new weights to avoid the trivial solution where the weight norm tends to infinity. We also require that the network output is unchanged when the neuron is added. Note that, due the latter, gradients of the existing weights are unchanged during growth and therefore we maximize the gradients on the new weights introduced.

$$\underset{\boldsymbol{W}_\ell^{\text{new}}, \boldsymbol{W}_{\ell+1}^{\text{new}}}{\arg\max} \left\| \mathbb{E}_D \left[ \frac{\partial L}{\partial \boldsymbol{W}_\ell^{\text{new}}} \right] \right\|_F^2 + \left\| \mathbb{E}_D \left[ \frac{\partial L}{\partial \boldsymbol{W}_{\ell+1}^{\text{new}}} \right] \right\|_F^2 \text{ s.t. } \begin{cases} \|\boldsymbol{W}_\ell^{\text{new}}\|_F, \|\boldsymbol{W}_{\ell+1}^{\text{new}}\|_F \le c \\ \boldsymbol{W}_{\ell+1}^{\text{new}} \boldsymbol{h}_\ell^{\text{new}} = 0 \end{cases} \quad (4)$$

**Motivation: Large gradients lead to large objective decrease** Consider running gradient descent on a function that is differentiable with a $\beta$-Lipschitz gradient. A classical result for this class of functions is that the decrease in objective function after one step of gradient descent with step-size $1/\beta$ is upper bounded by $L(\boldsymbol{W}_\ell) - \frac{\beta}{2}\|\nabla L(\boldsymbol{W}_\ell)\|^2$ (Nesterov, 2003). This upper-bound decreases as the norm of the gradient increases.

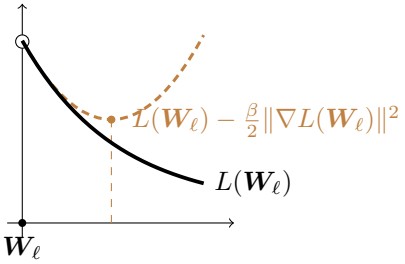

This observation implies that the larger the gradient, the larger the expected decrease (assuming a constant Lipschitz constant). Hence, interleaving gradient maximizing steps within training will lead to an improved decrease in later iterations. We use this observation in the context of growing neural networks and propose a method that interleaves the normal training process with steps that maximize the gradient norm by adding new units, which we now describe in detail.

**GradMax** The general maximization problem in eq. (4) is non-trivial to solve. However, we show that with some simplifying assumptions we can find an approximate solution using a singular value decomposition (SVD). Here we derive this solution using fully connected layers and share derivation for convolutional layers at Appendix B.

Consider 3 fully connected layers denoted with indices $\ell-1$, $\ell$ and $\ell+1$ and the following recursive definition:

$$\boldsymbol{z}_\ell = \boldsymbol{W}_\ell \boldsymbol{h}_{\ell-1} \quad (5)$$
$$\boldsymbol{h}_\ell = f(\boldsymbol{z}_\ell), \quad (6)$$

where subscripts denote layer-indices. Let $M_\ell$ denote the number of units in layer $\ell$, and $f$ the activation function. The vectors $\boldsymbol{z}_\ell, \boldsymbol{h}_\ell \in \mathbb{R}^{M_\ell}$ are pre-activations and activations respectively, with $\boldsymbol{h}_0 = \boldsymbol{x}$ being an input sampled from the dataset $D$. The entries of $\boldsymbol{W}_\ell \in \mathbb{R}^{M_{\ell-1} \times M_\ell}$ are the parameters of the layer.

When growing $k$ neurons the weight matrices $\boldsymbol{W}_\ell$ and $\boldsymbol{W}_{\ell+1}$ are replaced by $\boldsymbol{W}_\ell^+$ and $\boldsymbol{W}_{\ell+1}^+$ as defined in eqs. (2) and (3). We denote the pre-activations and activations of the new neurons with

$\boldsymbol{z}_\ell^{\text{new}}$ and $\boldsymbol{h}_\ell^{\text{new}}$. The gradients of the new weights can be derived using the chain rule:

$$\frac{\partial L}{\partial \boldsymbol{W}_\ell^{\text{new}}} = \left( f'(\boldsymbol{z}_\ell^{\text{new}}) \odot \boldsymbol{W}_{\ell+1}^{\text{new},\top} \frac{\partial L}{\partial \boldsymbol{z}_{\ell+1}} \right) \boldsymbol{h}_{\ell-1}^\top \tag{7}$$

$$\frac{\partial L}{\partial \boldsymbol{W}_{\ell+1}^{\text{new}}} = \frac{\partial L}{\partial \boldsymbol{z}_{\ell+1}} \boldsymbol{h}_\ell^{\text{new},\top}. \tag{8}$$

The simplifying assumptions that we will now make are that $\boldsymbol{W}_\ell^{\text{new}} = 0$ and that $f(0) = 0$ with gradient $f'(0) = 1$. Note that this guarantees that $\boldsymbol{W}_{\ell+1}^{\text{new}} \boldsymbol{h}_\ell^{\text{new}} = 0$, independent of the training data. Moreover, it simplifies the gradients to

$$\frac{\partial L}{\partial \boldsymbol{W}_\ell^{\text{new}}} = \boldsymbol{W}_{\ell+1}^{\text{new},\top} \frac{\partial L}{\partial \boldsymbol{z}_{\ell+1}} \boldsymbol{h}_{\ell-1}^\top \tag{9}$$

$$\frac{\partial L}{\partial \boldsymbol{W}_{\ell+1}^{\text{new}}} = 0, \tag{10}$$

which reduces our problem to

$$\underset{\boldsymbol{W}_{\ell+1}^{\text{new}}}{\arg\max} \left\| \boldsymbol{W}_{\ell+1}^{\text{new},\top} \mathbb{E}_D \left[ \frac{\partial L}{\partial \boldsymbol{z}_{\ell+1}} \boldsymbol{h}_{\ell-1}^\top \right] \right\|_F^2, \quad \text{s.t. } \left\| \boldsymbol{W}_{\ell+1}^{\text{new}} \right\|_F \le c. \tag{11}$$

The solution to this maximization problem is found in a closed-form by setting the columns of $\boldsymbol{W}_{\ell+1}^{\text{new}}$ as the top $k$ left-singular vectors of the matrix $\mathbb{E}_D \left[ \frac{\partial L}{\partial \boldsymbol{z}_{\ell+1}} \boldsymbol{h}_{\ell-1}^\top \right]$ and scaling them by $\frac{c}{\|(\sigma_1,...,\sigma_k)\|}$ (where $\sigma_i$ is the $i$-th largest singular value). In order to make a fair comparison between different methods we scale each initialization such that their norm is equal to the same value, i.e., mean norm of the existing neurons (similar to Liu et al. (2017)).

Note that it is feasible to also use the singular values to guide **where** and **when** to grow, since the singular values are equal to the value of the maximized optimization problem above. For example, neurons could be added when the singular values meet a certain threshold, and layers to grow could be chosen depending on which have the largest singular values. We leave this for future work.

**Non-linearities and normalization**   Note that our assumptions that $f(0) = 0$ and $f'(0) = 1$ apply to common activation functions such as rectified linear units (if zero is included in the linear part) and the hyperbolic tangent. Other functions could be adapted to fit this definition as well (e.g., by shifting the output of the logistic function by $-\frac{1}{2}$).

In fact our derivation holds for any $a$ such that $f'(0) = a, a \neq 0$. This means that batch normalization can also be used, since it meets these assumptions. However, care must be taken with activation functions (or their derivatives) that are unstable around zero. For example, batch normalization scales the gradients by $\frac{1}{\varepsilon}$ when all inputs are zero, which can lead to large jumps in parameter space (Lewkowycz et al., 2020, and Appendix C).

**GradMax-Optimized (GradMaxOpt)**   One can also directly optimize eq. (11) using an iterative method such as projected gradient descent. Our experiments show that this approach generally does not work as well as using the SVD, highlighting the benefit of having a closed-form solution.

However, if the outgoing weights are set to zero ($\boldsymbol{W}_{\ell+1}^{\text{new}} = 0$) instead of the incoming weights then the solution can no longer be found using SVD and direct optimization of eq. (4) could provide a solution. This could be preferable in some situations since it removes the constraints on the activation function. Moreover, it can avoid the unstable behaviour of functions such as batch normalization.

**Full gradient estimation**   Solving eq. (11) requires calculating the gradient $\frac{\partial L}{\partial \boldsymbol{z}_{\ell+1}} \boldsymbol{h}_{\ell-1}^\top$ over the full dataset. This can be expensive so in practical applications we propose using a large minibatch instead. If the minibatch gradient is sufficiently close to the full gradient then the top singular vectors are likely to be close as well (Stewart, 1998). We validate this experimentally in Section 4.2.

## 4 EXPERIMENTS

We evaluate gradient maximizing growing (GradMax) using small multilayer perceptrons (MLPs) and popular deep convolutional architectures. In Section 4.1 we focus on verifying the effectiveness

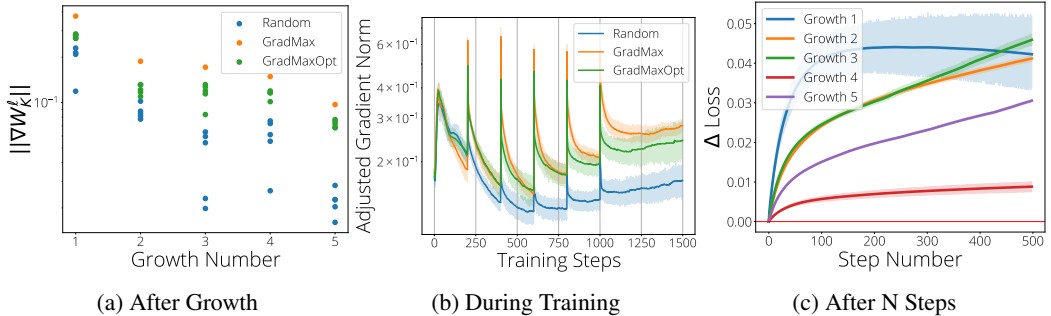

(a) After Growth          (b) During Training          (c) After N Steps

Figure 2: **(a)** We measure the norm of the gradients with respect to $W_\ell^{\text{new}}$ after growing a single neuron starting from checkpoints generated during the Random (Random) growth experiments. Since Random and GradMaxOpt are stochastic we repeat those experiments 10 times. **(b)** Gradient norm of the flattened parameters (both layers) throughout training. **(c)** Similar to (a), we load checkpoints from each growth step (Growth 1-5) and grow a new neuron using GradMax ($f_g$) and Random ($f_r$). Then we continue training for 500 steps and plot the difference in training loss (i.e., $L(f_r) - L(f_g)$). The confidence intervals are defined over 5 repetition of the experiment described above.

of GradMax and in Section 4.2 we evaluate GradMax on common image classification benchmarks and show the effect of different hyper-parameter choices.

We implement GradMax using Tensorflow (Abadi et al., 2015) and modify the implementation of the standard ReLU activation to output sub-gradient 1 at 0. In Appendix A we share the implementation details of GradMax and show how it can be used for growing new layers. We will open-source our implementation upon publication.

## 4.1 TEACHER-STUDENT EXPERIMENTS WITH MLPS

We have 3 main goals in this section. First, we empirically verify that the gradient norm is significantly increased using our method. Then, we verify our hypothesis that increasing the gradient norm at a given step improves the training dynamics beyond the growing step. Finally, we gather experimental evidence that networks grown with GradMax can achieve better training loss and generalization compared to baselines and other methods in a teacher-student setting.

**Teacher-Student task**  We use a teacher-student task (Lawrence et al., 1997) to compare different growing algorithms. In this setting a student network must learn the function of a given teacher network. Our teacher network $f_t$ consists of $m_i$ input nodes, $m_h$ hidden nodes and $m_o$ output nodes (denoted with $m_i : m_h : m_o$). We initialize weights of $f_t$ randomly from the range $[-1, 1]$ and then sample $N = 1000$ training points $D_x$ from a Gaussian distribution with zero mean and unit variance and pass it through $f_t$ to obtain target values $D_y$. With this training data we train various student networks ($f_s$) minimizing the squared loss between $f_s(D_x)$ and $D_y$.

A key property of this teacher-student setting is that when the student network has the same architecture as the teacher network, the optimization problem has multiple global minima with 0 training loss. Here we highlight results using fully connected layers alone. In Appendix D we show additional validation experiments as well as repeat experiments from this section with convolutions and batch normalization.

**Verifying GradMax**  In our initial experiments we use a teacher network of size $20 : 10 : 10$. All growing student networks begin with a smaller hidden layer of $20 : 5 : 10$. We grow the hidden layer to match the teacher architecture in 5 growth steps performed every 200 training steps. We also train two baseline networks with sizes $20 : 10 : 10$ (*Baseline-Big*) and $20 : 5 : 10$ (*Baseline-Small*) from scratch. After the final growth we perform 500 more steps resulting in 1500 training steps in total. We also run the version of GradMax in which we optimize eq. (11) directly using gradient descent (GradMaxOpt). We start with a random initialization and use the Adam optimizer. The weight matrix is scaled to the target norm $c$ after each gradient step.

In Figure 2a we show that in this setup GradMax is able to initialize the new neurons with a significantly higher gradient norm compared to random growing. The results for GradMaxOpt show that naive direct optimization of the gradient norm does not recover the solution found by GradMax

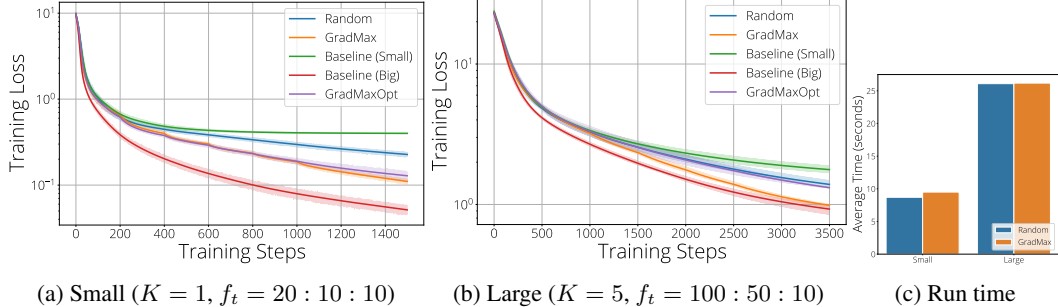

(a) Small ($K = 1$, $f_t = 20 : 10 : 10$)     (b) Large ($K = 5$, $f_t = 100 : 50 : 10$)     (c) Run time

Figure 3: Training curves averaged over 5 runs and provided with 80% confidence intervals. In both settings GradMax improves optimization over Random. **(right)** Run time of the growing algorithms.

using SVD. This highlights the benefit of having formulated a problem that can be directly solved. In Figure 2b we plot the total adjusted gradient norm[2] and observe that the larger gradient norm after growing persists for future training steps. Finally, in Figure 2c we plot the difference in training loss when GradMax is used for growing compared to randomly growing. We observe consistent improvements in the training dynamics which last for over 500 training steps. This supports our hypothesis that increasing the gradient norm accelerates training and leads to a greater decrease in the training loss for subsequent steps.

**Training Curves**    We plot the training curves for two different teacher networks in Figure 3. For Figure 3a we trained a teacher network of $20 : 10 : 10$ whereas Figure 3b uses a larger teacher network of size $100 : 50 : 10$. For the large teacher network setting we begin with student networks of size $100 : 25 : 10$ that are grown every 500 steps (first growth on step 500), adding 5 neurons at a time resulting for 5 growth steps. We train the grown networks for an additional 1000 iterations resulting in 3500 iterations in total. Further experimental details are shared in Appendix D.

In these experiments the initial student models require about half the number of FLOPS required by the teacher models. Therefore training and growing student models with a linear growing schedule costs only 75% of the FLOPS required to train *Baseline-Big*. The extra operations required by Grad-Max run faster than a single training step and therefore their cost is negligible as shown in fig. 3c. In all cases GradMax achieves lower training loss compared to the random and GradMax-Optimized (GradMaxOpt) methods. For the larger teacher network (Figure 3b), GradMax even matches the performance of a network trained from scratch (*Big-Baseline*). However for small teacher networks (Figure 3a), all growing methods fall short of matching the *Baseline-Big* performance similar to Berner et al. (2019); Ash & Adams (2020), in which the authors show that training the final network from scratch works better than warm-starting/growing an existing network. GradMax narrows down this important gap by maximizing gradients.

## 4.2    IMAGE CLASSIFICATION EXPERIMENTS

In this section we benchmark GradMax using various growing schedules and architectures on CIFAR-10, CIFAR-100, and ImageNet. GradMax can easily be applied to convolutional networks (see Appendix B for implementation details). Similar to our results in the previous section, we share baselines where we train the seed architecture (*Small-Baseline*) and the target architecture (*Big-Baseline*) from scratch without growing. As an additional baseline we implement a simpler version of Firefly which initializes new neurons by minimizing the training loss directly (without the extra candidates used by the original method). We refer to this baseline as Firefly-Optimized (Firefly-Opt). Firefly requires non-zero outgoing weights in order to optimize the loss. Therefore we initialize the outgoing weights for all methods to small random values ($\|\boldsymbol{W}_{\ell+1}^{\text{new}}\| = \varepsilon$ for $\varepsilon = 10^{-4}$). We use a batch size of 512 for ImageNet (128 for CIFAR) experiments and calculate the SVD using the same batch size used in training. At every growth iteration, we add new neurons to all layers with reduced initial width, proportional to their target widths. Here our goal is not to obtain state of art results, but to assess how the initialization of new neurons affects the final performance. All networks are trained for the same total number of steps, and thus *Baseline-Small* and the grown

---

[2]We divide the gradients by the training loss to factor out the linear scaling effect of the training loss on the gradient norms.

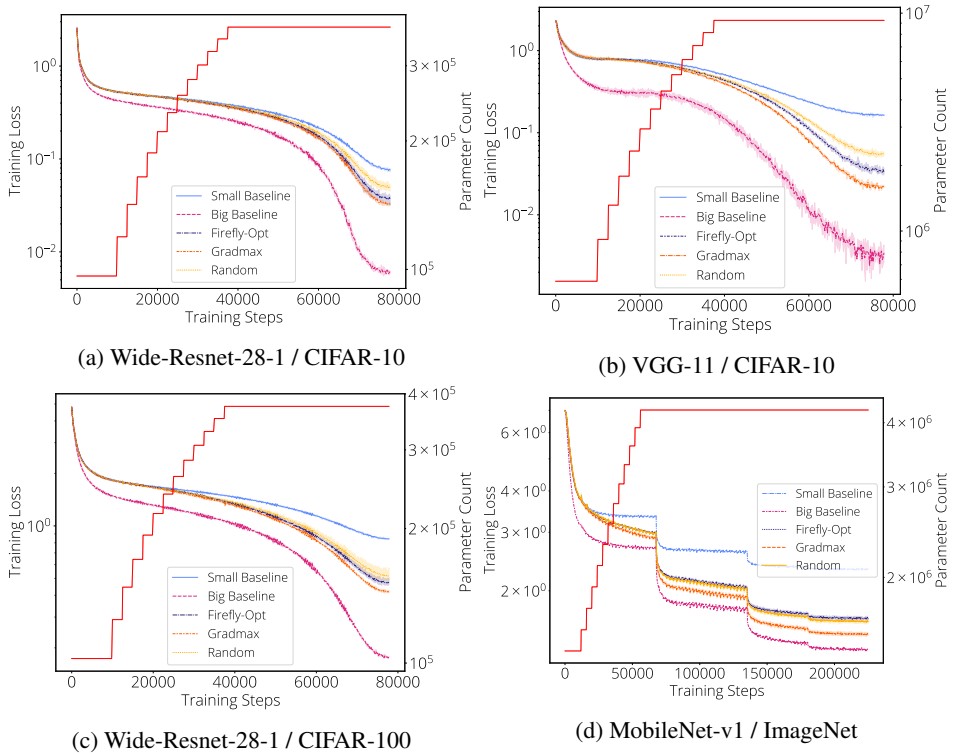

Figure 4: We plot training loss over time, each experiment is averaged over 3 experiments. Number of parameters of the network trained increases over time. Red lines indicate number of parameters over training.

| Dataset | Architecture | Baseline-S | Baseline-B | Random | Firefly | Gradmax |
|---------|-------------|-----------|-----------|--------|---------|---------|
| CIFAR-10 | WRN-28-1 | 89.9±0.3 | 92.9±0.2 | 90.6±0.2 | **90.8±0.3** | **91.1±0.1** |
| | VGG11 | 84.1±0.1 | 86.6±0.3 | 83.8±0.6 | **84.0±0.2** | **84.4±0.4** |
| CIFAR-100 | WRN-28-1 | 63.7±0.0 | 69.3±0.1 | **66.7±0.4** | 66.5±0.1 | **66.8±0.2** |
| ImageNet | Mobilenet-V1 | 55.0±0.0 | 70.8±0.0 | 66.9±0.3 | 66.4±0.1 | **68.6±0.2** |

Table 1: Test accuracy of different baselines and growing methods on different tasks. All results are averaged over 3 random seeds (used for training).

models require less FLOPS than *Baseline-Big* to train. We share our results in Table 1 and observe consistent improvements when GradMax is used.

**CIFAR-10/100** We perform experiments on CIFAR-10 and CIFAR-100 using two different architectures: VGG-11 (Simonyan & Zisserman, 2015) and WRN-28-1 (Zagoruyko & Komodakis, 2016). For both architectures we reduce number of neurons in each layer by 1/4 to obtain the seed architecture (width multiplier=0.25). For the ResNet architecture we only shrink the first convolutional layer at every block to prevent mismatch when adding skip-connections. We train the networks for 200 epochs using SGD with a momentum of 0.9 and decay the learning rate using a cosine schedule. In both experiments we use a batch size of 128 and perform a small learning rate sweep to find learning rates that give best test accuracy for baseline training. We find 0.1 for Wide-ResNet and 0.05 for VGG to perform best and use the same learning rates for all different methods. In Figures 4a to 4c we plot the training loss for different growing strategies and share the test accuracy in Table 1.

**ImageNet** We grow MobileNet-v1 architectures on ImageNet using a seed architecture of width 0.25 (i.e. all layers have one forth of the channels). The MobileNet-v1 architecture contains depthwise convolutions between each convolutional layers. When growing convolutional layers, we initialize the depth-wise convolution in between to identity. Training loss for this settting is shared in Figure 4d.

| BN | Inverse | Baseline-S | Baseline-B | Random | Firefly | Gradmax(-Opt) |
|----|---------|------------|------------|--------|---------|---------------|
| ✗ | ✗ | 89.9±0.3 | 92.9±0.2 | 90.6±0.2 | 90.8±0.3 | 91.1±0.1 |
| ✗ | ✓ | | | 92.1±0.2 | 92.2±0.2 | 92.4±0.1 |
| ✓ | ✗ | 90.2±0.3 | 93.4±0.1 | 92.9±0.1 | 92.9±0.1 | 93.0±0.1 |
| ✓ | ✓ | | | 92.8±0.1 | 92.8±0.2 | 92.9±0.2 |

Table 2: Average test accuracy when growing WRN-28 on CIFAR-10 with batch normalization and outgoing weights set to zero. *BN* refers to batch normalization and *inverse* indicates that we set the outgoing weights to zero. When the outgoing weights are set to zero, we use GradmaxOpt.

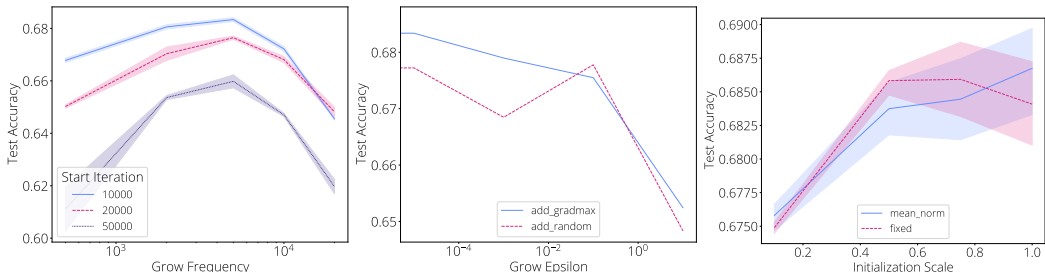

Figure 5: We grow MobileNet-v1 networks during the ImageNet training to investigate the effect of **(left)** growing schedule, **(center)** $\varepsilon$, and **(right)** scale used in initialization on the test accuracy.

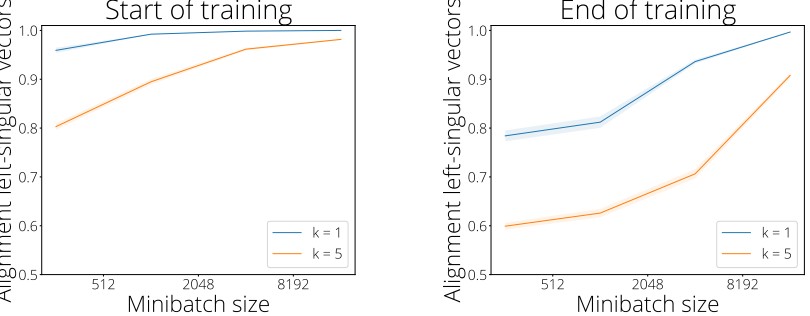

Figure 6: The alignment of the top-$k$ left-singular vectors for WRN-28-1 during the CIFAR-10 training. The alignment is calculated between the full gradient and minibatches of varying sizes. We do not apply random cropping or flips to the inputs. A total of 10 experiments are run and confidence intervals of 95% are plotted.

**Batch Normalization Results and Inverse Formulation**   We perform our main set of experiments using networks without batch norm and setting the incoming weights to zero. However, in some cases using batch norm or setting outgoing weights to zero might be useful. Table 2 shows the results for such alternatives when growing residual networks on CIFAR-10. For this particular setting we observe that batch norm has limited effect on results, however we observe consistent improvements when outgoing weights are set to zero. We use GradmaxOpt in this setting and observe improvements over both random and Firefly.

**Hyperparameters**   In Figure 5 we show the effect of various hyperparameters on the performance of GradMax when growing MobileNet-v1 on ImageNet. First on Figure 5-left we compare different growing schedules. Growing neurons early in the training and growing every 5000 step achieves the best results. Subfigure Figure 5-center shows the effect of $\varepsilon$ used in our experiments. As expected, larger values of $\varepsilon$ (and therefore not preserving the output of the network during the growing) harm the final test accuracy. Finally, Figure 5-right shows the effect of the scale used when initializing the new neurons. We compare normalizing the scales using the mean norm of existing neurons (*mean*) to using no normalization (*fixed*) and observe no visible different in performance. We observe relatively robust performance for values larger than 0.5 and therefore simply set the scale to 0.5 in all of our experiments.

**Effect of Minibatch Size** To validate that we can approximate the SVD of the full gradient with the SVD of a large minibatch we calculate both and plot the alignment of the left-singular vectors. We define alignment as the absolute value of the Frobenius trace of the two matrices of singular vectors, normalized with $\frac{1}{k}$. The plots in Figure 6 show that at the beginning of training the singular vectors strongly align. At the end of training the gradients are less correlated and hence the alignment decreases. However, it is still a reasonable estimate.

## 5 RELATED WORK

Cascade-Correlation (CasCor) learning architecture (Fahlman & Lebiere, 1990; Shultz & Rivest, 2000), to our knowledge, is the first algorithm that grows a neural networks during training. CasCor adds new layers that consists of single neurons that maximize the correlation between centered activations and error signals at every growth step by resulting in a very deep but narrow final architecture. Platt (1991) allocates new neurons whenever the per-sample error is high. Fukumizu & Amari (2000) show that local minima can turned into saddle points by adding neurons. Chen et al. (2016) proposed splitting neurons and Wei et al. (2016) presented a more general study of performance-preserving architecture transformations. Elsken et al. (2017) apply these transformations to NAS. Wen et al. (2019) proposed growing neurons and discovered that it is best to grow layers early on. Lu et al. (2018) also add new layers, but they sparsify the resulting network. *Convex neural networks* frames the learning of one-hidden layer neural networks as an infinite-dimensional convex problem (Bengio et al., 2006; Rosset et al., 2007; Bach, 2017) and then approximates the solution by an incremental algorithm that adds a neuron at each step.

Kilcher et al. (2019) show that the scaling of the outbound weights can be optimized with respect to gradient norm when the original network is at local minima. The solution is given by the closed form using a matrix of outbound partial derivatives. Liu et al. (2019) propose an algorithm for splitting neurons. For each neuron it computes an approximate reduction in loss if this weight were cloned and adjusted by a given offset. The solution to this problem is given by the eigendecomposition of a "splitting matrix". The splitting is most effective when the network is very close to convergence. Calculating the splitting matrix is expensive so in Wang et al. (2019) the authors propose a faster approximation. Wu et al. (2020b) expand the set of splitting directions which helps avoid local minima in architecture space. In the context of manifold learning, Vladymyrov (2019) also proposed to grow the number of dimensions of the embedding manifold in order to escape local minima. Similar to our approach, new parameters are found in a closed form.

Gradient based growth criteria is used in the context of growing sparse connections (Liu et al., 2017; Dettmers & Zettlemoyer, 2019; Evci et al., 2020; ab Tessera et al., 2021; Evci et al., 2022) and when initializing neural networks (Dauphin & Schoenholz, 2019). Most relevant to our work is NeST (Liu et al., 2017) which aims to maximize gradients when growing sparse neurons by wiring neurons with highest correlations. However their method is limited to growing a single neuron and modifies the output of existing neurons due to non-zero initialization on both sides.

## 6 DISCUSSION

**Limitations** GradMax requires activation functions to map zero to zero and have non-zero derivative at the origin. Some activation functions, such as radial basis functions, don't satisfy these requirements. We studied fully connected layers and convolutional layers, but did not consider the combination of the two or other architectures such as transformers.

**Future work** We are looking to address these limitations in future work. Moreover, in this work we didn't address the questions as to when and where networks should be grown, although we outlined ways in which our method could be adapted to do so. Our method could also be a useful network morphism to be used as part of a more complex NAS method. These are further avenues for exploration.

## 7 CONCLUSION

In this work we presented GradMax, a new approach for growing neural networks that aims to maximize the gradient norm when growing. We verified our hypothesis and demonstrated that it provides faster training in a variety of settings.

ACKNOWLEDGEMENTS

We like to thank members of the Google Brain team for their useful feedback. Specifically we like to thank Mark Sandler, Andrey Zhmoginov, Elad Eban, Nicolas Le Roux, Laura Graesser and Pierre-Antoine Manzagol for their feedback on the project and the preprint.

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

## A    IMPLEMENTATION DETAILS

GradMax requires the calculation of the quantity $\frac{\partial L}{\partial \boldsymbol{z}_{\ell+1}} \boldsymbol{h}_{\ell-1}^{\top}$ (eq. (9)). The user could perform this outer product manually, but in most machine learning frameworks the following approach is easier to implement.

We note that when $\boldsymbol{W}_{\ell}^{\text{new}} = 0$ and $f(0) = 0$ we can write

$$\boldsymbol{W}_{\ell+1}^{\text{new}} f(\boldsymbol{W}_{\ell}^{\text{new}} \boldsymbol{h}_{\ell-1}) = \boldsymbol{W}_{\ell}^{\text{aux}} \boldsymbol{h}_{\ell-1}, \quad \boldsymbol{W}_{\ell}^{\text{aux}} \in \mathbb{R}^{M_{\ell-1} \times M_{\ell+1}} \tag{12}$$

and

$$\boldsymbol{z}_{\ell+1} = \boldsymbol{W}_{\ell+1} \boldsymbol{h}_{\ell} + \boldsymbol{W}_{\ell}^{\text{aux}} \boldsymbol{h}_{\ell-1} . \tag{13}$$

Note that $\boldsymbol{W}_{\ell}^{\text{aux}} = 0$. The advantage of this formulation is that

$$\frac{\partial L}{\partial \boldsymbol{W}_{\ell}^{\text{aux}}} = \frac{\partial L}{\partial \boldsymbol{z}_{\ell+1}} \boldsymbol{h}_{\ell-1}^{\top} . \tag{14}$$

Hence GradMax can easily be implemented in any framework with automatic differentiation by temporarily introducing this auxiliary matrix $\boldsymbol{W}_\ell^{\text{aux}}$ in order to calculate $\frac{\partial L}{\partial \boldsymbol{z}_{\ell+1}} \boldsymbol{h}_{\ell-1}^\top$. After the columns of $\boldsymbol{W}_{\ell+1}^{\text{new}}$ are calculated using SVD, the matrix $\boldsymbol{W}_\ell^{\text{aux}}$ is replaced with the new layer $(\boldsymbol{z}_{\ell+1} = \boldsymbol{W}_{\ell+1}\boldsymbol{h}_\ell + \boldsymbol{W}_{\ell+1}^{\text{new}} f(\boldsymbol{W}_\ell^{\text{new}}\boldsymbol{h}_{\ell-1}))$.

**Growing Layers**   Matrix decomposition point of view provides a novel insight into the growing problem since it can be applied to any 2 layers and can help us grow new one between the two. Note that, in our initial derivation we studied the problem of adding new neurons to an existing layer. Now we can choose any 2 layers $k$ and $\ell$ and create a new layer between them through growing. If there is already a layer between the two (i.e. $k = \ell - 2$ or $k = \ell + 2$), then the new neurons are appended to the existing layers.

## B   CONVOLUTIONS

The derivation of GradMax for fully connected layers can readily be extended to convolutional layers:

$$\frac{\partial L}{\partial \boldsymbol{W}_\ell^{\text{new}}} = \left( f'(\boldsymbol{z}_\ell^{\text{new}}) \odot \boldsymbol{W}_{\ell+1}^{\text{new}} *^\top \frac{\partial L}{\partial \boldsymbol{z}_{\ell+1}} \right) * \boldsymbol{h}_{\ell-1} \tag{15}$$

$$= \boldsymbol{W}_{\ell+1}^{\text{new}} * \left( \frac{\partial L}{\partial \boldsymbol{z}_{\ell+1}} * \boldsymbol{h}_{\ell-1} \right) \tag{16}$$

$$\frac{\partial L}{\partial \boldsymbol{W}_{\ell+1}^{\text{new}}} = \frac{\partial L}{\partial \boldsymbol{z}_{\ell+1}} * \boldsymbol{h}_\ell^{\text{new}} \tag{17}$$

$$= 0 \tag{18}$$

where $*$ is a 2D convolution and $*^\top$ is the transposed 2D convolution (i.e., the gradient of the convolution).

As before, the required gradient is in practice most easily calculated by introducing an auxiliary convolutional layer such that $\frac{\partial L}{\partial \boldsymbol{W}_\ell^{\text{aux}}} = \frac{\partial L}{\partial \boldsymbol{z}_{\ell+1}} * \boldsymbol{h}_{\ell-1}$. We let

$$\boldsymbol{z}_{\ell+1} = \boldsymbol{W}_{\ell+1} * \boldsymbol{h}_\ell + \boldsymbol{W}_\ell^{\text{aux}} * \boldsymbol{h}_{\ell-1} \, . \tag{19}$$

where $\boldsymbol{W}_\ell^{\text{aux}} = 0$ are filters of the appropriate size. Namely, if $\boldsymbol{W}_\ell$ are filters of size $(h_\ell, w_\ell)$ with $i_\ell$ input channels and $\boldsymbol{W}_{\ell+1}$ contains filters of size $(h_{\ell+1}, w_{\ell+1})$ with $o_{\ell+1}$ output channels, then $\boldsymbol{W}_\ell^{\text{aux}}$ will have filters of size $(h_\ell + h_{\ell+1} - 1, w_\ell + w_{\ell+1} - 1)$ with $i_\ell$ input and $o_{\ell+1}$ output channels.

We now have the equivalent of equation 11 to solve for convolutional layers:

$$\arg\max_{\boldsymbol{W}_{\ell+1}^{\text{new}}} \left\| \boldsymbol{W}_{\ell+1}^{\text{new}} * \mathbb{E}_D \left[ \frac{\partial L}{\partial \boldsymbol{z}_{\ell+1}} * \boldsymbol{h}_{\ell-1}^\top \right] \right\|_F^2, \quad \text{s.t. } \left\| \boldsymbol{W}_{\ell+1}^{\text{new}} \right\|_F \le c \, . \tag{20}$$

We express the 2D convolution as a matrix multiplication (Vasudevan et al., 2017) using the *im2col* method. This means that $\boldsymbol{W}_{\ell+1}^{\text{new}}$ is flattened to a matrix of size $\mathbb{R}^{o_{\ell+1} \cdot h_{\ell+1} \cdot w_{\ell+1} \times k}$. The matrix $\mathbb{E}_D \left[ \frac{\partial L}{\partial \boldsymbol{z}_{\ell+1}} * \boldsymbol{h}_{\ell-1}^\top \right]$ is of size $\mathbb{R}^{i_\ell \times o_{\ell+1} \times h_\ell + h_{\ell+1} - 1 \times w_\ell + w_{\ell+1} - 1}$ but is turned into a larger matrix of size $\mathbb{R}^{o_{\ell+1} \cdot h_{\ell+1} \cdot w_{\ell+1} \times i_\ell \cdot h_\ell \cdot w_\ell}$ by extracting the appropriate patches (depending on the filter sizes, padding, and strides) which are flattened and then concatenated. SVD can then be readily applied to this reformulated problem and the resulting top-$k$ left-singular vectors can be reshaped to form the filters of the new layer.

## C   BATCH NORMALIZATION

Batch normalization scales the activations to have unit variance. This makes the incoming weights scale-free (Arora et al., 2018; Li & Arora, 2020), which is usually a desirable property. However, it can be disruptive when growing neurons. For the first step after growing, batch normalization keeps the zero activations and scales the gradients with $\frac{1}{\epsilon}$. After the first gradient step the incoming weights will be non-zero, resulting in non-zero activations which will be scaled to unit variance. This can

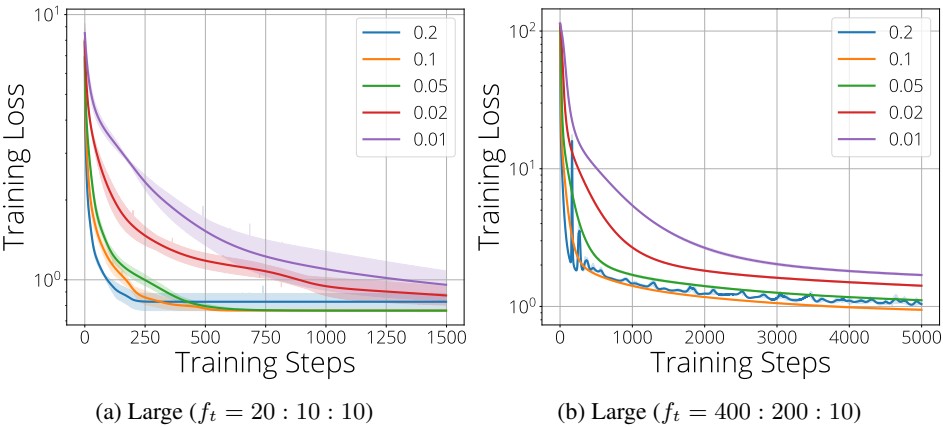

(a) Large ($f_t = 20 : 10 : 10$)          (b) Large ($f_t = 400 : 200 : 10$)

Figure 7: Effect of learning rate on the optimization speed and quality. We repeat each experiment 3 times with a different seed and report the average values with 80% confidence intervals.

## D    ADDITIONAL EXPERIMENTAL DETAILS FOR STUDENT-TEACHER EXPERIMENTS

**Learning rate.** For both experiments we perform a small hyper parameter search over the learning rate. In Figure 7 we share learning curves using different learning rates that range between $0.2$ and $0.01$. Higher learning rates bring faster convergence, however best results are obtained with the second highest learning rate equal to $0.1$. To avoid confounds related to the optimizer, we restrict the experiments to gradient descent with constant learning rate.

**Correlation between singular values and the final decrease.** We know from the theory that the norm of the gradient is proportional to the singular values given by the solution of equation 11. In other words, solution corresponding to the top singular values should give the largest norm of the gradient for new weights and therefore should have the best training dynamics in the long run. In Figure 8 we demonstrate this on a simple Student/Teacher set up as we used in the main paper. We first train own network without any growing (blue curve on the top graph). We then grow the network at different stages of optimization using any of the five singular values and let the network converge to some value. As we can see, darker colors corresponding to the larger singular values generally converge faster and correspond to the lower values of the objective function.

At the lower portion of the figure, we quantify this result more precisely by repeating the experiment five times and computing the Pearson correlation coefficient between the vector of singular values and the loss decrease some iterations after growing. There $x$-axis represents how many iterations have passed since the growth and the $y$-axis shows the correlation between the loss values at that iteration and the singular values. The lower the values the more (anti-)correlated the values are and, therefore, the more we can trust the GradMax algorithm. The iteration it always negative and gets worse as we iterate. However, if we use GradMax at the later stages of the algorithm close to convergence, the correlation holds for a while.

**Student-Teacher Experiments with Convolutional Networks.** Similar to the experiments in Figure 2a, here we show the results of the Student-Teacher task on a convolutional model. As visible on Figure 9, the same conclusions can be made for this case as well. GradMax achieves the best norm of the gradient after each growth. Interestingly, the gradient norm is somewhat larger for the Random growth, but in Figure 10 we see that GradMax achieves better results overall.

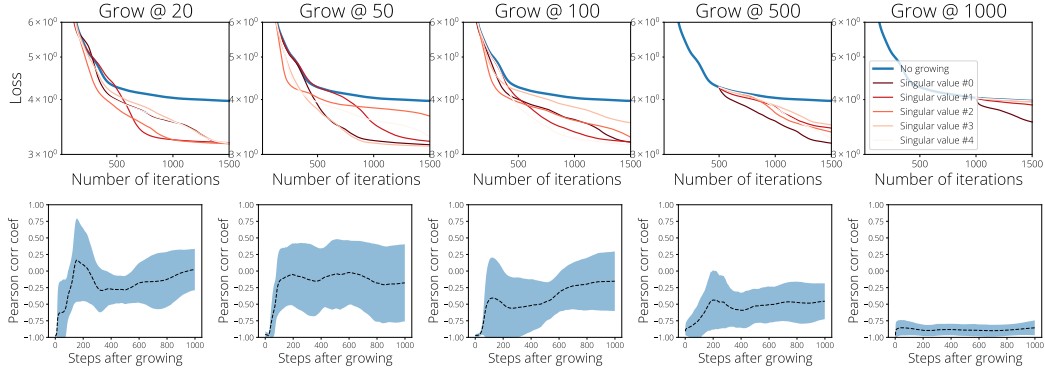

Figure 8: Long-term effect of growing. *Top:* Growing once with one of 5 singular values starting at different iteration (20th, 50th, 100th, 500th or 1000th). Blue curve represents no growing. *Bottom:* The correlation between the singular values and the loss function decrease after a certain growing iteration.

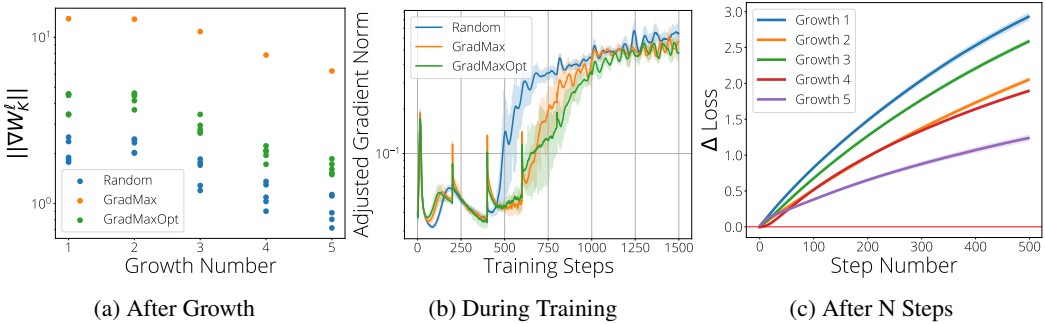

(a) After Growth       (b) During Training       (c) After N Steps

Figure 9: **(a)** In this plot we load checkpoints from each growth step generated during the Random experiments. We grow a single neuron and measure the norm of the gradients with respect to $\boldsymbol{W}_\ell^{new}$. Since Random and GradMaxOpt are stochastic we repeat those experiments 10 times. GradMax provides the maximum gradient norm **(b)** We measure the norm of flattened parameters throughout the training. GradMax improves gradient norm over Random. **(c)** Similar to (a), we load checkpoints from each growth step and grow a new neuron using GradMax ($f_g$) and Random ($f_r$). Then we continue training for 500 steps and plot the difference in training loss (i.e. $L(f_r) - L(f_g)$). All experiments are repeated 5 times.

**Student-Teacher Experiments with Batch Norm**    Similar to the MLP and convolution cases, Figures 11c and 12 provide results for the GradMax performance for the model with BatchNorm.

**Setting both incoming and outgoing weights to zero**    In these experiments we set both side of the new neuron to zero and initialize the bias to 1. This non-zero bias provides unit activation, which provides non-zero gradients for the outgoing weights. After the first step, outgoing weights become non-zero and learning takes off. Results in Figure 13a show that this approach performs worse than GradMax.

**Effect of start iteration and initial width**    Here, first, we look at the effect of growing new neurons during different parts of the training. We repeat small the student-teacher experiments in Section 4.1 using GradMax, but adjust the iteration the first neuron is grown. We grow 5 neurons in total with 200 step intervals. We also adjust the training steps after the last growth so that all experiments match in training FLOPs. Results are presented in Figure 13b. We observe that growing early in the training works best, which highlights a different role for growing algorithms than what is presented in the literature (Fukumizu & Amari, 2000): Neural networks can be grown during training with the goal of improving training dynamics. Our results show the potential of such an approach. Next, in Figure 13c, we study the effect of initial network size while keeping total FLOPs used during train-

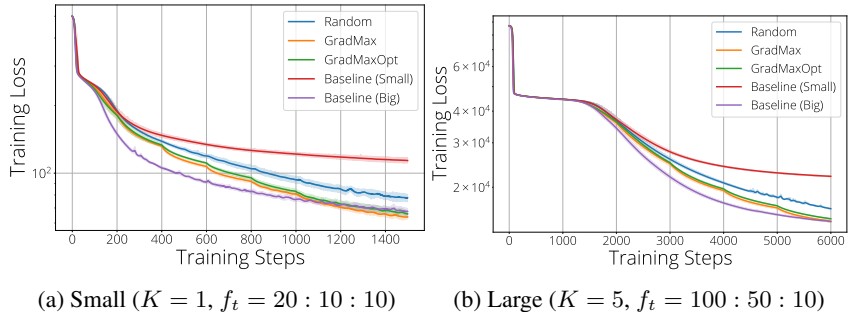

(a) Small ($K = 1$, $f_t = 20 : 10 : 10$)  (b) Large ($K = 5$, $f_t = 100 : 50 : 10$)

Figure 10: Training curves averaged over 5 different runs and provided with 80% confidence intervals. In both settings GradMax significantly improves optimization over Random. Split-based methods seems to cause some instability in large network settings causing frequent jumps in the training objective.

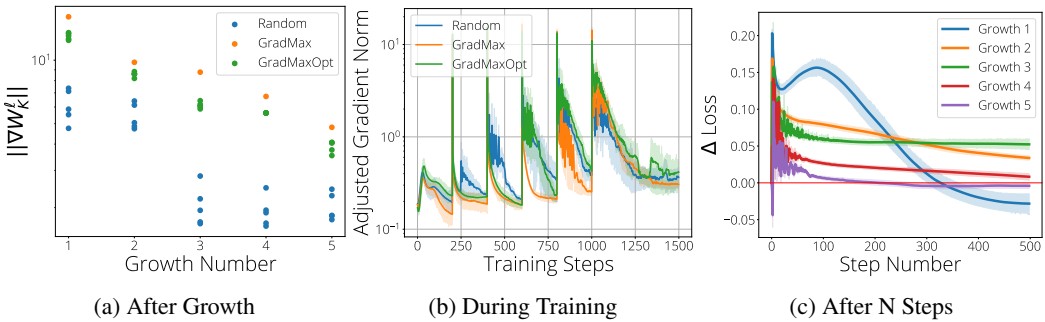

(a) After Growth  (b) During Training  (c) After N Steps

Figure 11: **(a)** In this plot we load checkpoints from each growth step generated during the Random experiments. We grow a single neuron and measure the norm of the gradients with respect to $W_{\ell}^{new}$. Since Random and GradMaxOpt is stochastic we repeat those experiments 10 times. GradMax provides the maximum gradient norm **(b)** We measure the norm of flattened parameters throughout the training. GradMax improves gradient norm over Random. **(c)** Similar to (a), we load checkpoints from each growth step and grow a new neuron using GradMax ($f_g$) and Random ($f_r$). Then we continue training for 500 steps and plot the difference in training loss (i.e. $L(f_r) - L(f_g)$). All experiments are repeated 5 times.

ing constant. As expected, starting with a larger student model helps optimization and obtains better training loss for both GradMax and Random, while GradMax consistently achieves better results than Random in all settings.

**Adding More Neurons** In our student-teacher experiments we stop growing new neurons when the student model matches the teacher model in terms of number of neurons. In Figure 13d we grow the student model beyond the size of the teacher model and observe with around 50% more neurons (1.5x), the student model exceeds the baseline (big) performance.

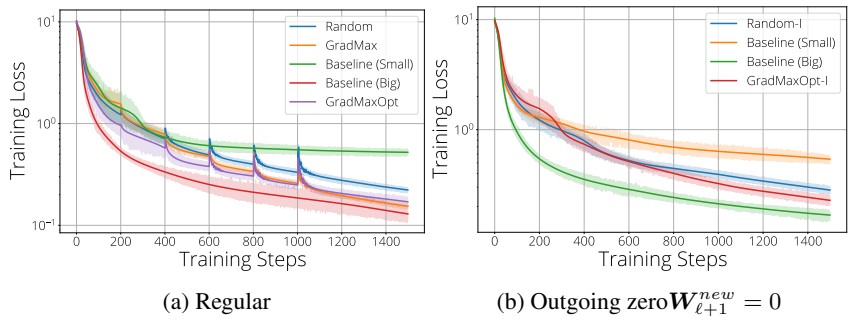

(a) Regular  (b) Outgoing zero$\boldsymbol{W}_{\ell+1}^{new} = 0$

Figure 12: Training curves averaged over 5 different runs and provided with 80% confidence intervals. In both settings GradMax significantly improves optimization over Random. Split-based methods seems to cause some instability in large network settings causing frequent jumps in the training objective.

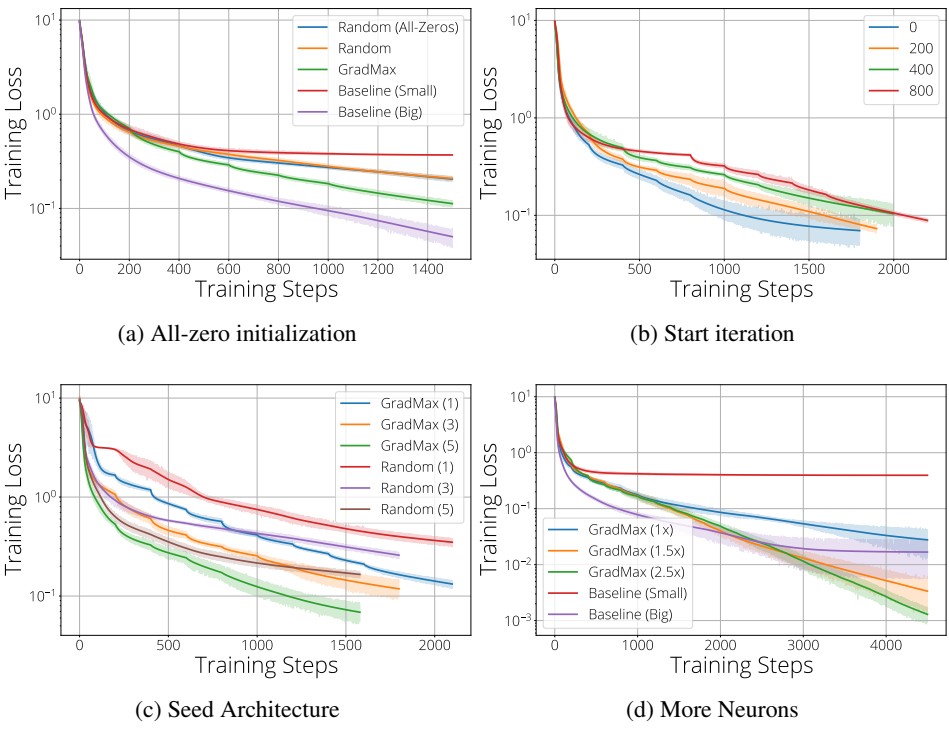

(a) All-zero initialization  (b) Start iteration

(c) Seed Architecture  (d) More Neurons

Figure 13: **(a)** We compare setting all-weights to zero (and having unit bias) to our approach of setting only the incoming weights to zero (e.g. Random and GradMax). **(b)** Effect of growing during different parts of the training for GradMax. Labels correspond to the iteration when the first neuron is grown. We grow 5 neurons in total with 200 step intervals. Training steps are adjusted so that all runs have same amount of FLOPs. **(c)** Effect of initial student network size. As before we adjust total number of steps so that the total training FLOPs match for all experiments. Numbers in parentheses represent the size of the hidden layer at initialization. We grow neurons starting from first training iteration. **(d)** Growing beyond teacher network shows that with larger capacity, grown networks exceed the baseline performance.

