# OpenReview forum: "GradMax: Growing Neural Networks using Gradient Information"
_ICLR.cc/2022/Conference — ICLR 2022 Poster_

### Official Review · Reviewer_LS3D · 2021-11-01

**Correctness:** 2
**Technical Novelty And Significance:** 2
**Empirical Novelty And Significance:** 2
**Recommendation:** 5
**Confidence:** 3

**Main Review:**

The paper is well written and easy to follow. Although I am not familiar with the literature, the idea seems novel and the experiments on common image datasets are comprehensive and well-executed. The improvements seem marginal but are consistent throughout the paper.

Nevertheless, the motivation of the paper is not well explained and needs more mathematical and empirical justification. As far as I understand, the authors want to introduce (neurons and therefore) weights such that the overall gradient of the loss wrt to all the weights is minimized. I find this idea interesting but not well reflected in the considered optimization problem, equation 4. Since the introduction of the new weights also change the backward/gradient computation of layers k < l, it could be that although optimization problem 4 is well solved, the overall gradient decreases - which would go against the motivation/ method proposed.
Please elaborate on this point and explain why this can't happen. The teacher-student experiments are not sufficient because no weights before the newly introduced W_l^new exist and I cant see anything about this for the larger models (with batchnorm).

On top of this, this could be turned around and part of the method: How can we add new neurons such that we change/improve the backward pass in way that improves the gradient norm of the whole network e.g. by backward computation that circumvents vanishing gradients. In general, the motivation of the paper should be much more investigated and could be made much stronger.

Another important point that is missing in my oppionin is the careful handling of batchnorm layers.
Scaling the norm of the (newly added) weights has no effect in the forward pass but scales the norm of the gradient in the backward pass, see e.g. (https://openreview.net/forum?id=rkxQ-nA9FX, https://arxiv.org/pdf/1910.07454.pdf).
This goes hand in hand with your motivation but is never discussed for models when batchnorm is used. Please elaborate on this and show that your motivation and the proposed method still make sense with batchnorm layers.

If the authors manage to convince me / improve my mathematical understanding and intuition of what the method is actually doing, I would consider raising my score.


**Summary Of The Paper:**

The paper proposes a method to grow neural networks by adding neurons in a specific way. While requiring that the forward pass stays unchanged, the authors want to initialize the new weights such that the gradient wrt to these weights is maximized. They show small but consistent improvements compared to the baselines although the proposed method seems considerably more complex than baselines, as far as I can see.


**Summary Of The Review:**

Not sufficient and convincing evidence that the motivation of the paper make sense and is met in practice.

---

> ### Author Response · Authors · 2021-11-16
> **Authors' Response**
>
> We are glad that the reviewer agrees that the idea is novel, that our experiments are comprehensive and well-executed, and that the improvements are consistent. We thank the reviewer for pointing out the related work on batch normalization and their question on our motivation/formulation. We believe there is a misunderstanding on the effect of growth on the existing neural network, which we clarify below and also incorporate in the paper. We hope this resolves the remaining questions about our work.
>
> > As far as I understand, the authors want to introduce (neurons and therefore) weights such that the overall gradient of the loss wrt to all the weights is minimized.
>
> No, the goal is to add weights such that the norm of the gradient with respect to the new weights is maximized (see equation 4).
>
> > Since the introduction of the new weights also change the backward/gradient computation of layers k < l, it could be that although optimization problem 4 is well solved, the overall gradient decreases
>
> This is actually not the case. Note that we set either the incoming or outgoing weights of the new neurons to zero. This guarantees that the gradient signal propagated through the new neurons is also set to zero and will not affect the backward computation through the lower layers of the network. We added a sentence in the text that clarifies this.
>
> > How can we add new neurons such that we change/improve the backward pass in way that improves the gradient norm of the whole network
>
> The only way that the new neurons could change the gradients with respect to the old weights in our setting, is if both the new incoming and outgoing weights are non-zero. This would change the output of the function, which would violate the constraint in equation 4.
>
> Although this could of course be a reasonable thing to do, it raises many questions that lie outside of the scope of this paper: How do we increase the norm of the gradients without increasing the loss? How do we avoid making our network unstable by increasing the norm of the gradients once we are near the optimum? For these reasons we follow the general assumption that is made in most of the research on growing, and leave the function output (approximately) unchanged. See also figure 5b that justifies this assumption by experimentally showing the negative effects of having non-zero incoming weights.
>
> > Scaling the norm of the (newly added) weights has no effect in the forward pass but scales the norm of the gradient in the backward pass
>
> We discussed this in the text in the section “Non-linearities and normalization”, but did not have space to elaborate further. Batch normalization does indeed present some difficulties because it scales the gradients by 1/ε on the backward pass. This means that the first step taken after growing can be very large. After this first update, the incoming weights are no longer zero and regardless of the learning rate on the new weights, the activations of the new neuron jump from zero to unit variance. We observed using small values for the outgoing weights to be useful to reduce the effect of this jump to the rest of the network. Alternatively, as we propose in the text, batch normalization can be used with GradMaxOpt without needing any adaptations. We added this discussion to the paper as Appendix C citing the papers proposed.

---

> ### Author Response · Authors · 2021-11-28
> **Remaining Concerns**
>
> Approaching to the end of the discussion period, we wanted to follow-up on this thread. Does our response addresses reviewers' concern on motivation, intuition and handling of batch normalization? We are happy to address any remaining concerns in the remaining time. If they are all addressed would the reviewer consider raising their scores?

---

### Official Review · Reviewer_igLv · 2021-11-02

**Correctness:** 3
**Technical Novelty And Significance:** 4
**Empirical Novelty And Significance:** 4
**Recommendation:** 6
**Confidence:** 4

**Main Review:**

The paper is somewhat confusingly written, and it could be better organized. For example, the loss curves are given for the training set in Figure 5 for convnets on CIFAR-10, CIFAR-100, and Imagenet. This is appropriate, given that the method is intended to accelerate training by focusing on the gradients. However, the performance of these networks on the test set are in Table 1, on the previous page. Table 1 should be moved so it shows up *after* the training curves. I give some more suggestions below.

Strengths:

The work provides a novel, practical approach to growing networks.

The method is cost-effective, as solving the SVD problem is fast. Its motivation and derivation are presented (mostly) clearly.

This provides a new alternative to Neural Architecture Search (NAS).

The method applies to modern convolutional networks.

Weaknesses, with concrete, actionable feedback

I find the writing confusing at times.

The second paragraph in Section 4 mentions three goals. It doesn't mention that you are going to test this on convolutional networks (Section 4.2). This introductory paragraph should include that as a fourth goal, as it is an introduction for the whole section 4, not just 4.1. This reader at least, was surprised when I got to Section 4.2, as I was beginning to think you weren't going to try anything relevant to deep convolutional networks. It is best to set expectations.

Figure 3C displays the relative difference in loss reduction between GradMax and randomly-initialized units added at the same point in training. Another baseline would be to add units with 0 weights (both input and output), with a bias such that they have some small activation, and thus get incorporated into the network.

In general, while the experiments are somewhat systematic, they start in the middle somehow. Would you always start with a network of some size? Can you apply the method from the beginning of training, starting one unit or channel per layer? In a practical setting, how would you decide to stop adding units? What happens to these networks if you keep adding units?

*Minor issues/typos/wording suggestions*

First paragraph of Section 3: Change:

We also require that the network output before and after the growth remains unchanged.

to:

We also require that the network output is unchanged when the neuron is added.

In the paragraph after equation 4, you say that:

"A classical result for this class of functions is that the *decrease in objective function* after one step of gradient descent with step-size 1/β is upper bounded by L(Wl) − β2 ∥∇L(Wl)∥2 (Nesterov, 2003). This is an expression that decreases as the norm of the gradient increases." [emphasis added]

This says that the *decrease of the loss* gets smaller as the gradient gets bigger, so obviously, if that were true, you would want a *smaller* gradient, which makes no sense. What I think you meant to say was that the *loss* is upper bounded by this expression, so you want the gradient to get bigger. Is that right?

Bottom of page 4, regarding Figure 2. Here, you don't say (in the text, anyway), that this is WRN 28-1 (or describe what that network is), what the task is, or the training set. Have you defined k at this point?

Section 4.1, second paragraph:

teacher-student settings ->

teacher student setting

The sentence: "The baselines Baseline-Small and Baseline-Big are trained using the fixed architectures 20:5:10 and 20:10:10." is confusing because you define them after you use their name.

Better: We define two baseline networks trained from scratch. Baseline-Big is the size of the final network, 20:10:10. Baseline-Small is the size of the initial student network, 20:5:10.

Same page, second line from the bottom: the SVD -> SVD.

Page 6, middle:

You measure the norm of the parameters? Don't you mean the norm of the gradient?

This experiments is repeated 5 times ->

This experiment is repeated 5 times.

Training curves paragraph: When do you *start* adding nodes? 500 training steps?

Section 4.2, first sentence ends with CIFAR-10. You should end it with "CIFAR-10, CIFAR-100, and ImageNet." (no need to add "2012").

Next sentence:

Note that GradMax can be applied... ->

GradMax can easily be applied...

line or two down:

by minimizing the training loss directly ->

by minimizing the training loss with a single new neuron

For both architectures we start using a width multiplier of 0.25 for our seed architecture. ->

For both architectures we start with a layer 1/4 the original width for our seed architecture.

Then you say you apply this to the first layer of resnet, but you don't say what layer you apply it to for VGG-11.

Similarly, for MobileNet-v1, you don't say what layer you shrink.

Table 2: What dataset is this for? What networks?

Bottom of page 6:

We observe relatively robust for values...->

We observe relatively robust performance for values...

Related work: You don't reference the canonical example of growing networks: Scott Fahlman's Cascade-Correlation networks from 1992 or so. I notice that there are recent video lectures posted online of Scott talking about cascade correlation and deep learning. I also found a blog post entitled: "Cascade-Correlation, a Forgotten Learning Architecture"! So you aren't the only one...

Page 9: real-basis functions -> radial basis functions?

Various places:

fig X -> Figure X or Fig. X

table y -> Table y

subfigure 4b -> Figure 4(b)

For the larger network, GradMax even matches...->

For the larger teacher network (Figure 4(b)), GradMax even matches...

However for small teacher networks, all growing methods...

However for small teacher networks (Figure 4(a)), all growing methods...

"This highlights an important gap to fill in future research." Ok, but in the current environment, most people care about big networks!

Bottom of page 6: You mention the number of flops to train - it would probably be good to squeeze in a bar plot of flops of the various methods. This is a big selling point of the method.

Bottom of page 13: If the norm of the gradient is inversely proportional to the singular values, you would want to use the smallest singular values, but obviously, you want the biggest ones, right? Can you clear this up for me?



**Summary Of The Paper:**

This paper presents a new method (as far as I know) of growing neural networks. This can be viewed as an approach to finding the meta-parameter corresponding to network size. The main idea is to add units such that they maximize the norm of the gradient. This is suggested to be superior to methods that grow by adding units that minimize the loss, as maximizing the gradient will lead to learning benefits in subsequent training. With some simplifying assumptions, this turns out to be an optimization problem solvable by SVD. Although it is a full batch method, using a large minibatch size provides a practical algorithm. The results show that the intuition is correct: The change in loss over following epochs continues to be larger than if randomly-initialized units are added (see Figure 3(c)). The networks are able to grow to the size of a reference network trained from scratch, and achieve performance that is not far from this standard in some cases. It is not expected here that growing the network will achieve equivalent performance to a properly-sized network trained from scratch, but it uses less compute due to starting with smaller networks. The approach is first tested in fully-connected, shallow networks, but generalizes to convolutional networks, including resnets, hence, it is a practical approach to growing networks.

I have read the authors' response and am happy to raise my score a notch. I also think this is a worthwhile direction for research.

**Summary Of The Review:**

This is a novel idea (as far as I know) for growing networks, with a good intuition: add units so that they maximize the gradient, rather than minimize the loss. This has good effects on the learning dynamics. Unfortunately, the writing and organization dampens enthusiasm for the paper, and the experiments seem a bit haphazard. I think this paper could be improved considerably by addressing these issues.

---

> ### Author Response · Authors · 2021-11-16
> **Authors' Response**
>
> We truly appreciate the reviewer's detailed review. We incorporated the feedback given, which we believe helped improve our work greatly, we thank the reviewer for that. Here is our response.
>
> > The second paragraph in Section 4 mentions three goals. It doesn't mention that you are going to test this on convolutional networks (Section 4.2).
>
> Thanks for noticing this mistake. This paragraph was intended to be for Section 4.1. We moved the existing paragraph to Section 4.1 and added a short introduction that covers the entire section.
>
> > Figure 3C displays the relative difference in loss reduction between GradMax and randomly-initialized units added at the same point in training. Another baseline would be to add units with 0 weights (both input and output), with a bias such that they have some small activation, and thus get incorporated into the network.
>
> We implemented this initialization and compared it with Random growth and GradMax in Figure 13a (Appendix D):  *In these experiments we set both sides of the new neuron to zero and initialize the bias to 1.  This non-zero bias provides unit activation, which provides non-zero gradients for the outgoing weights. After the first step, outgoing weights become non-zero and learning takes off. Results in Figure 13a show that this approach performs worse than GradMax.*
>
> > Would you always start with a network of some size? Can you apply the method from the beginning of training, starting one unit or channel per layer?
>
> Yes, indeed GradMax can be used to even grow layers (0 neurons). We added a discussion on this to Appendix A. In Fig. 13c we show that GradMax maintains its advantage over random when the initial network has 1,3 or 5 neurons. Similarly we added Fig. 13b to show the benefits of growing during early training (similar to results in Fig. 5a).
>
> >  In a practical setting, how would you decide to stop adding units? What happens to these networks if you keep adding units?
>
> Adding more units increases the cost of the resulting neural network. In practice, network size is decided according to the cost requirements of the application. When growing, similarly, one can stop according to the requirements. With more units added, optimization accelerates and training loss decreases further. We added Fig. 13d to demonstrate this, where we kept adding new neurons (upto 2.5x size of the teacher network) and observed that using around 50% more neurons(1.5x), the student model exceeds the baseline (big) performance.

---

> > ### Author Response · Authors · 2021-11-16
> > **Authors' Response (Cont.)**
> >
> > ### Minor Issues
> > We thank the reviewer again for taking the time to share a long list of minor suggestions/typos. We fixed/incorporated them in the main text.
> > > What I think you meant to say was that the loss is upper bounded by this expression, so you want the gradient to get bigger. Is that right?
> >
> > Indeed, we updated the text: `This upper-bound decreases as the norm of the gradient increases.`
> >
> > > Bottom of page 4, regarding Figure 2.
> >
> > We moved Figure 2 to the experimental section (now Figure 6).
> >
> > > You measure the norm of the parameters? Don't you mean the norm of the gradient?
> >
> > Thanks for noticing the typo, `norm of the gradient is proportional to the singular values`. This can be seen in Eq. 11, assuming a fixed norm for W_{\ell+1}^{new}.
> >
> > > Training curves paragraph: When do you start adding nodes? 500 training steps?
> >
> > yes, when we grow every n steps; the first growth is done on nth step. We updated the text.
> >
> > > For both architectures we start with a layer 1/4 the original width for our seed architecture. Then you say you apply this to the first layer of resnet, but you don't say what layer you apply it to for VGG-11. Similarly, for MobileNet-v1, you don't say what layer you shrink.
> >
> > For all image classification experiments we reduce number of neurons in *every* layer. In literature the 'width multiplier` term is used to indicate this. Similarly when we grow, we grow neurons at every layer. We updated the text to make this clear.
> >
> > > Table 2: What dataset is this for? What networks?
> >
> >  WRN-28 / CIFAR-10: updated the caption.
> >
> > > Related work: You don't reference the canonical example of growing networks:
> > Scott Fahlman's Cascade-Correlation networks from 1992 or so.
> >
> > We had Cascade-Correlation Learning Architecture in our initial literature review, but forgot to add it. Thanks for pointing out this important work. We added a short comparison between GradMax and CasCor to the related work.
> >
> > > "This highlights an important gap to fill in future research." Ok, but in the current environment, most people care about big networks!
> >
> > We agree on the importance of the scaling of machine learning methods, and growing neural networks has great potential to reduce the cost of long-running large scale projects as presented in Fig. 4 of [1]. We updated our text to clarify our point at the end of Section 4.1: `However for small teacher networks (Figure 3a), all growing methods fall short of matching the *Baseline-Big* performance similar to Berner et al. (2019); Ash & Adams(2020),  in which the authors show that training the final network from scratch works better than warm-starting/growing an existing network. GradMax narrows down this important gap by maximizing gradients.`
> >
> > > Bottom of page 6: You mention the number of flops to train - it would probably be good to squeeze in a bar plot of flops of the various methods. This is a big selling point of the method.
> >
> > Thank you, we added this barplot (Fig3c).
> >
> > > Table 1 should be moved so it shows up after the training curves.
> >
> > We updated the location of Table-1 so that it comes after Fig. 4.
> >
> > [1] https://cdn.openai.com/dota-2.pdf

---

> ### Author Response · Authors · 2021-11-28
> **Remaining Concerns**
>
> Approaching to the end of the discussion period, we like to thank again the reviewer for their detailed review and helping us improving our paper. Did our response address reviewers' concern on organization and experiments? We updated our submission addressing reviewers' comments and hope the reviewer can find time to look at it. If reviewers' concerns are all addressed, would the reviewer consider raising their score further and support acceptance?

---

### Official Review · Reviewer_LUg4 · 2021-11-02

**Correctness:** 3
**Technical Novelty And Significance:** 3
**Empirical Novelty And Significance:** 3
**Recommendation:** 6
**Confidence:** 3

**Main Review:**

**Pros:**

+ This paper is well-written and neat. The motivation and the method are clear.

+ The method is simple but somewhat novel, I like simple ideas.

**Cons:**

- It seems many prior works (as described in section 2) are missing in experiments.

- Can authors list the time complexity of GradMax and GradMaxOpt?



**Summary Of The Paper:**

In this paper, the authors propose the GradMax algorithms, in order to add neurons that will cause better optimization results in latter iterations.

**Summary Of The Review:**

My weakness is listed above.

----
### Post rebuttal

After reading the author's response, I feel my questions are addressed. I will increase my score to 6.

---

> ### Author Response · Authors · 2021-11-16
> **Authors' Response**
>
> We agree that novel and simple ideas are the best! We think conferences such as ICLR should have more of those, instead of papers with complex algorithms providing incremental improvements to ImageNet. Which is why we ask the reviewer to kindly reconsider their score and give a sign that novel & simple counts more than top1 error on benchmarks -- We think the weaknesses and improvements pointed out by the reviews did help improve the paper's quality enough to warrant inclusion into a conference that aims to promote novelty and creativity.
>
> > It seems many prior works (as described in section 2) are missing in experiments.
>
> Growing methods that split existing neurons have some important limitations as pointed out in the main text: `1) It creates redundant neurons and small changes required to break symmetry cause changes in the output distribution; (2) It can't be used for growing new layers as it requires existing neurons to begin with.` Furthermore, the baseline method we compared our method against `Firefly` obtains better results than these methods and therefore adding splitting methods would lead to same conclusions.
>
> > Can authors list the time complexity of GradMax and GradMaxOpt?
>
> Strictly speaking both methods have the same complexity: O(nmk) where n and m are the sizes of the layers, and k is the number of neurons being added. An exact comparison between the time complexities of GradMax and GradMaxOpt is difficult to characterize since the convergence of top-k SVD using Lanczos/LOBPCG depends on the gap between singular values.
>
> The important thing to note is that we only require a single forward and backward to calculate the n ⨉ m gradient matrix. This means that for a large neural network, the remaining O(nmk) computation for SVD (GradMax) or gradient descent (GradMaxOpt) is almost negligible and amortised during regular training. We added Fig3c, to demonstrate this amortisation.
>
> [firefly] https://arxiv.org/abs/2102.08574

---

### Official Review · Reviewer_1m2b · 2021-11-03

**Correctness:** 3
**Technical Novelty And Significance:** 3
**Empirical Novelty And Significance:** 2
**Recommendation:** 6
**Confidence:** 3

**Main Review:**

[Strengths]
- The idea of the proposed weight initialization method, maximizing the gradient norm of the new weights, is intuitively reasonable. Also, the proposed SVD-based algorithm seems to be technically sound.
- The advantage of the proposed GradMax against random initialization is shown through the numerical experiments.

[Weaknesses]
- Although GradMax works well when growing neurons in an existing layer, the reviewer thinks that GradMax is not applicable when adding a new layer to networks. Perhaps, it may be possible by adding a layer together with skip connections.
- Although adding neurons gradually in model training can reduce the total training cost, the performance degradation compared to training the big model from scratch (Baseline (Big)) is not negligible.
- This paper only focuses on the weight initialization when adding new neurons to the network. However, as the author pointed out, when and where new neurons should be added is important in practice. Also, when we should stop the neuron growing is important in terms of finding a good network structure. The proposed GradMax should be combined with an architecture search method and evaluated to show the effectiveness in the practical use case.

**Summary Of The Paper:**

This paper proposes a weight initialization method when growing neural networks. The key idea is to initialize the new neuron's weights to maximize their gradient norm. The proposed initialization is realized by the singular value decomposition (SVD) under some assumptions. The effectiveness of the proposed gradient maximizing growth (GradMax) is verified on the simple artificial tasks and image classification tasks in the scenario for increasing the number of neurons.

**Summary Of The Review:**

The proposed weight initialization method is reasonable when adding neurons in model training. However, the experimental evaluation assumes a limited situation. The effectiveness of the proposed method in practical usage is not apparent.

---

> ### Author Response · Authors · 2021-11-16
> **Authors' Response**
>
> We thank the reviewer for sharing their concerns. We agree that practical usage is an important aspect of research. Though our work has some important signals on this front (see ImageNet/MobileNet results), this is not the main focus of our work. Practical algorithms sometimes take multiple research projects to reach and our work should be considered as a first step towards this. Below we respond to the individual points.
>
> > Although GradMax works well when growing neurons in an existing layer, the reviewer thinks that GradMax is not applicable when adding a new layer to networks. Perhaps, it may be possible by adding a layer together with skip connections.
>
> It's straight-forward to add new layers with Gradmax. We added a discussion of how to use GradMax for growing new layers to the Appendix A. To see how this could work, consider the Matrix decomposition formulation of Gradmax, used in our implementation and explained in Appendix A. This formulation provides a novel insight into the growing problem since it can be applied to any 2 layers, thus can help us grow a new layer between existing ones. With this formulation, we can choose any 2 layers (let's say $k$ and $\ell$) and grow a new auxiliary layer between them initialized at zero. This zero-initialized layer is then decomposed into new neurons. If there is no layer between $k$ and $\ell$, then these new neurons make up the new layer. If there is, they are appended to the existing layer. More details can be found in Appendix A.
>
> > Although adding neurons gradually in model training can reduce the total training cost, the performance degradation compared to training the big model from scratch (Baseline (Big)) is not negligible.
>
> Suboptimal performance of growing a neural network (compared to training the final network from scratch) is observed in previous work [1, 2]. Understanding and improving the optimization of a growing network is an active area of research and our work narrows the gap observed. We would also like to note that, as shown in Fig4 of [1], growing/warm-starting has significant cost benefits (8x faster experimentation and thus significantly cheaper). Furthermore, training from scratch might not be possible in certain settings. While the performance degradation is not negligible, for some of today's largest models the total training costs are very large (e.g. GPT-3). In those cases, our method could offer a valuable alternative despite this degradation. We think that researchers and practitioners should at least be aware that this alternative exists so they can make this decision for themselves.
>
> > This paper only focuses on the weight initialization when adding new neurons to the network. However, as the author pointed out, when and where new neurons should be added is important in practice. Also, when we should stop the neuron growing is important in terms of finding a good network structure. The proposed GradMax should be combined with an architecture search method and evaluated to show the effectiveness in the practical use case.
>
> We thank the reviewer for their interest and suggestion. It is common for research papers to study one aspect of learning in depth [3], possibly due to space constraints.  "When" and "where" to grow new neurons can change from application to application. However, a better initialization method ("how") should be applicable to all different growing scenarios.
>
> Although we are limited by space, we do believe that GradMax frames growth in such a way that it can naturally answer questions about “where” and “when”, simply by looking at the singular values of the “growing matrices” and growing whenever/wherever a singular value meets a certain threshold.
>
> - [1] https://cdn.openai.com/dota-2.pdf
> - [2] https://arxiv.org/pdf/1910.08475.pdf
> - [3] https://proceedings.mlr.press/v9/glorot10a/glorot10a.pdf

---

> > ### Comment · Reviewer_1m2b · 2021-11-27
> > **Comments after the authors' response**
> >
> > Thank you for your responses. I understood how to add a new layer by the proposed GradMax. Although I want to see the experimental result when the number of layers gradually increases by GradMax, I recognize the generality of the proposed method. I expect that a detailed discussion about the limitation and potential of GradMax, including the combination of architecture growing methods, will be added to the paper.
> > In summary, I am satisfied with the authors' responses. Therefore, I will raise my score.

---

### Author Response · Authors · 2021-11-16
**Meta-comment**

We thank the reviewers for their time. We are glad that the reviewers agree that the idea is simple (Reviewer LUg4), intuitive (Reviewer LS3D), novel (Reviewer igLv) and practical for growing (Reviewer igLv) We are also happy that the Reviewer igLv noticed that our experiments are comprehensive and well-executed, and that the improvements are consistent.

*Any common theme emerged from reviews that need to be addressed generally?*

We also like to highlight following FAQ from the [ICLR reviewers guide](https://iclr.cc/Conferences/2021/ReviewerGuide#faq)

>Q: If a submission does not achieve state-of-the-art results, is that grounds for rejection?

>A: No, a lack of state-of-the-art results does not by itself constitute grounds for rejection. Submissions bring value to the ICLR community when they convincingly demonstrate new, relevant, impactful knowledge. Submissions can achieve this without achieving state-of-the-art results.

Below we are going to answer each individual reviewers’ feedback. We will be happy to continue the discussion if the reviewers have any additional questions after that.

---

### Decision · Program_Chairs · 2022-01-20

**Decision:**

Accept (Poster)

**Comment:**

This paper looks into growing neural networks, and finds an improved approach to the initialisations of new layers, viz by maximising the gradient norm.  Simple, straightforward, neat, and no good reason to reject.  It will benefit those who are using growing NNs.